# Analysis of Thermal and Roughness Effects on the Turbulent Characteristics of Experimentally Simulated Boundary Layers in a Wind Tunnel

**DOI:** 10.3390/ijerph19095134

**Published:** 2022-04-23

**Authors:** Giuliano Demarco, Luis Gustavo Nogueira Martins, Bardo Ernst Josef Bodmann, Franciano Scremin Puhales, Otávio Costa Acevedo, Adrian Roberto Wittwer, Felipe Denardin Costa, Debora Regina Roberti, Acir Mércio Loredo-Souza, Franco Caldas Degrazia, Tiziano Tirabassi, Gervásio Annes Degrazia

**Affiliations:** 1Programa de Pós Graduação em Engenharia Mecânica, Universidade Federal de Santa Maria, Santa Maria 97105-900, Brazil; 2Departamento de Física, Universidade Federal de Santa Maria, Santa Maria 97105-900, Brazil; lgnm.sm@gmail.com (L.G.N.M.); franciano.puhales@ufsm.br (F.S.P.); otavio@ufsm.br (O.C.A.); fdenardin@unipampa.edu.br (F.D.C.); debora@ufsm.br (D.R.R.); tiziano.tirabassi@gmail.com (T.T.); gervasiodegrazia@gmail.com (G.A.D.); 3Laboratório de Aerodinâmica das Construções, Universidade Federal do Rio Grande do Sul, Porto Alegre 91501-970, Brazil; bardo.bodmann@ufrgs.br (B.E.J.B.); acir@ufrgs.br (A.M.L.-S.); fcdegrazia@yahoo.com.br (F.C.D.); 4Facultad de Ingeniería, Universidad Nacional del Nordeste, Resistencia 3500, Argentina; arwittwer@gmail.com; 5Campus Alegrete, Universidade Federal do Pampa UNIPAMPA, Alegrete 97546-550, Brazil

**Keywords:** wind tunnel experiments, thermal and roughness turbulent effects, convective boundary layer, turbulent energy spectra

## Abstract

The aim of this paper is to analyse the thermal effects in a wind tunnel experiment to simulate the planetary boundary layer (PBL). Experiments were performed in the wind tunnel of the Laboratory of Constructions Aerodynamics at the Federal University of Rio Grande do Sul, Porto Alegre, Rio Grande do Sul State, Brazil. This wind tunnel is a closed return low-speed wind tunnel specifically designed for dynamic and static studies on civil construction models. As a novelty, one of the experimental sections of the wind tunnel was equipped with a metal sheet with Peltier elements coupled to it. In other words, thermal effects generating new flow patterns become feasible and open pathways to compare wind tunnel simulations to those in the PBL. Furthermore, measurements obtained with the smooth floor of the wind tunnel were repeated under the same conditions with the addition of the roughness in the floor, and the mechanical turbulence generated by the surface roughness significantly amplified the exchange of momentum and heat between the regions located in vertical direction of the wind tunnel boundary layer. In the presence of turbulent heat flux near the surface, thermal effects contribute to the increase of the turbulence intensity. Turbulent energy spectra for flow velocities and different heights were obtained using the Hilbert–Huang transform method, and the observed convective turbulence energy spectra behavior reproduced those measured in an unstable surface PBL.

## 1. Introduction

Wind tunnel experiments are widely used to simulate reduced-scale scenarios of real-world interest, including PBL wind flows and their usage in wind power issues, among others [1,2]. The planetary boundary layer is the lowest part of the atmosphere that interacts directly with the surface. The turbulent properties of the planetary boundary layer control the surface fluxes of momentum, heat and mass between the atmosphere and the underlying surface. Therefore, a detailed understanding of the turbulent physical processes in this vertical region is of fundamental importance in applications in meteorology, wind enginnering and in atmospheric numerical models. The planetary boundary layer has a nearly two-dimensional structure, the larger the eddies, the more horizontal they become. However the turbulence in the planetary boundary layer generated by different forcing mechanisms originating from thermal (positive turbulent heat flux) and mechanical (wind shear and terrain roughness) can be considered a fully developed three-dimensional turbulence [3]. Small-scale turbulence, far from initial conditions and at high Reynolds numbers, tends to reach a statistically stationary state [4]. This assumption, for fully developed three-dimensional turbulence, is not valid for two-dimensional turbulence and other problems with an inverse cascade of energy, i.e., from small to large scale [5]. Considering these arguments, it is possible to establish a direct comparison of the wind tunnel flow with atmospheric boundary layer flows. However, most wind tunnel simulations are idealizations that may not be able of including effects measured in field experiments, such as vertical thermal fluxes due to the presence of solar radiation and other thermal sources [6,7,8]. In general, the wind flows under various mechanical and thermal conditions make field experiments difficult to reproduce under comparable conditions, leading studies to rely on wind tunnel experiments to establish conditions comparable to real-scale scenarios. In turbulent wind flows, physically relevant quantities are horizontal and vertical wind speeds, vertical temperature, pressure profile, and humidity distribution, which are subject to fluctuations due to non-linear flow dynamics and, on the other hand, due to complex boundary layer conditions [9]. Regarding wind tunnel experiments, it can be seen in the literature that mechanical and thermal forcings are considered to simulate realistic PBL [10,11].

A range of research has reported experimental results on neutral boundary layer flow in wind tunnels [12,13,14,15], and some of these studies referred to part-depth simulations of the atmospheric boundary layer and even the lower part of the boundary layer [16,17]. A critical aspect to consider in this type of physical simulation is the low value reached by the Re number of the reduced scale model concerning the atmospheric boundary layer itself [18]. The effects of surface roughness and roughness length estimation are also central aspects of the comparative analysis of this type of neutral PBL flow [19,20]. Moreover, the spectral analysis of these turbulent flows and the determination of suitable spectral models to compare simulated boundary layer flows in wind tunnels with atmospheric data are essential in small-scale experimental studies [21,22,23].

Based on these discussions, the present study includes thermal effects in wind tunnel experiments that simulate an atmospheric boundary layer. To achieve this, boundary layer flows were physically modeled in the “Joaquim Blessmann” wind tunnel of the Federal University of Rio Grande do Sul, Porto Alegre, State of Rio Grande do Sul, Brazil. This facility is a closed-return wind tunnel specifically designed for boundary layer experimental studies, and static and dynamic structural tests on civil construction models. To carry out this work, a sector of the wind tunnel test section was equipped with a metal sheet and Peltier elements attached to it. Peltier elements transport heat by electric currents and heat or cool the floor of the experimental section depending on the polarity. Thus, heating the floor simulates the effects of solar radiation. In other words, thermal effects generating new flow patterns become feasible and open pathways to compare wind tunnel simulations to planetary boundary layer conditions [24]. The planetary surface boundary layer turbulence is sustained by friction stress and vertical heat flux, while the higher regions of the planetary boundary layer, due to the Earth’s rotation effect, are influenced by Coriolis forces [25]. It is important to mention that in the wind tunnel experiments performed in this study no rotation effects are considered and therefore many phenomena induced by rotation are not simulated.

Furthermore, we present and discuss the turbulent energy spectrum of the velocity fluctuations obtained with different mean incident wind velocities. To do this, measurements of the velocity fluctuations were made at different heights. Smooth wind tunnel floor measurements were repeated under the same conditions with floor roughness. The time series of the longitudinal velocity component is determined with hot-wire anemometry and analyzed using the Hilbert–Huang transform method.This new spectral analysis uses an adaptive basis of functions that allows the low frequency region of turbulent energy spectra to be represented at a detailed resolution. This approach has been employed to analyze observed atmospheric complex signals [26,27]. Therefore, the novelty in this study is to use the Hilbert–Huang analysis procedure to derive the turbulent energy spectra from experimental data obtained under controlled flow conditions in a wind tunnel. Identifying frequency windows associated with the distinct functionalities of turbulent degrees of freedom is essential when studying turbulence. For instance, spectral peak frequencies provide spatial and temporal scales of the energy content of eddies responsible for dispersing scalar and vector quantities in a turbulent flow. Given the above, a test table was developed with Peltier elements to obtain a controlled heat flux, generating positive temperature gradients to carry out the vertical temperature gradient experiments.

In this study, both neutral and convective wind tunnel boundary layers are considered. A convective boundary layer occurs when there is a positive turbulent heat flux induced by a temperature difference between the surface and adjacent air. A neutral boundary layer means a type of boundary layer characterized by a zero heat flux state throughout its vertical depth. The remainder of the paper is as follows. Section 2 addresses the methodology describing the wind tunnel experiment. Section 3 presents and discusses the experimental results. For the conclusion, a general discussion of the main obtained results is presented.

## 2. Methodology

The wind tunnel experimental area has a length/height ratio greater than 10 and dimensions of 1.30×0.90×9.32 m (width × height × length). The maximum wind speed in this chamber, with a smooth and uniform flow, is 42 m s${}^{-1}$ (150 km h${}^{-1}$). The analysis and comparison of flows with different velocities between 1 and 5 m s${}^{-1}$ for neutral and convective stability regimes are reported herein, where the convective condition is driven by a temperature gradient of roughly 0.1 K m${}^{-1}$. Statistical parameters of turbulence are obtained in frequency domain based on the flow velocity fluctuation time series. Furthermore, the vertical temperature gradient profiles of the flow were analyzed.

With respect to the basic wind tunnel performance, mean velocity distribution across the working section indicates deviations of 2% from mean value out of the boundary layers. The mean values and standard deviations of the turbulence intensity for measurement points out of the boundary layers are 0.37% and 0.17%. These values and other wind tunnel technical specifications are reported by Blessmann [28].

### 2.1. Experimental Description

Measurements were conducted with a hot-wire anemometer probe (DANTEC 55P11 [29]) and CTA System Data Acquisition System—DANTEC-Stream-Line Pro (Dantec Dynamics—DK-2740 Skovlunde, Denmark). The calibration system allows measuring the characteristics of turbulent flows subject to temperature gradients due to temperature-dependent calibration. The high time resolution allows to measure fluctuations above 250 kHz and the spatial resolution allows eddies to be solved with spatial scales of the order of the Kolmogorov scale (∼1 mm). An automatic calibrator—Stream-Line Pro Automatic Calibrator (Dantec Dynamics—DK-2740 Skovlunde, Denmark) was used to calibrate hot-wire probes in airflow at very low speeds of the order of $10^{-1}$ m s${}^{-1}$ and up to speeds as high as $10^{2}$ m s${}^{-1}$. This calibrator is ideal as it compensates for variations in ambient temperature. The calibration interval for measurements with low wind speeds was in the 0 to $10^{1}$ m s${}^{-1}$ range. Moreover, the probe is located in a jet with a uniform velocity profile during the calibration process. This apparatus is connected to a computer via USB or Ethernet, and the calibration process is controlled by the DANTEC software StreamWare Pro (for further details, see [30]).

The experiments were performed on a turntable M-II of the wind tunnel “Prof. Joaquim Blessmann” (Figure 1). Physical simulations of the boundary layer were contemplated by two types of flows (a boundary layer under neutral and unstable stability conditions). While the isothermal flow obtained a neutral boundary layer, the convective boundary layer was generated by the positive heat flux, and the metal surface of the M-II table was heated from below by 12 Peltier elements (type TE-127-1.4-1.5). Peltier elements were distributed over two circles on an aluminum sheet of 3 mm thickness and with a diameter of 800 mm. The Peltier elements were installed on a smaller circle with four elements and a larger one with eight elements in order to cool or heat segments of the plate with the same area. Each element had coupled a heat sink with a square section equal to the geometry of the Peltier element (40 mm × 40 mm × 4 mm). A more detailed description of the table with Peltier elements is given in reference [30]. The velocity and temperature profiles were simultaneously measured using the same anemometer system installed in experimental Turntable M-II (Figure 1). A miniature wire probe was used for velocity and turbulence measurements, simultaneously with a specific temperature sensor. Additionally, thermocoupled sensors on the surface of table II were used to control the convective boundary layer stationarity. The measuring points were obtained for 5, 10, 15, 20, 25, 30, 40, 50, 60, 70, 80, 90, 100, 150, 200, 250 and 300 mm, respectively. A transverse system was employed and the measurements were performed point-to-point. The positions indicated refer to the direction normal to the tunnel floor and were measured from the floor level of the test section.

Four velocity magnitudes, corresponding to the freestream velocity of 0.9, 1.3, 1.9 and 3.9 m s${}^{-1}$ of the wind tunnel, were analyzed. At each measurement position, a time series of 368,640 values was obtained with a sampling frequency of 2048 Hz, which corresponded to a 180 s duration for each data record. Simultaneously, the records corresponding to the velocity fluctuation variance and temperature at each measurement point were evaluated. Signal post-processing did not include low-pass and high-pass filter.

The values for the friction velocity ($u^{*}$) were estimated using the mean velocity data and Prandtl’s known wall law. For the configuration without roughness, the values $u^{*}=0.05,\;0.07,\;0.10$ and 0.21 m/s were obtained for the cases $U_{\infty }=0.9,\;1.3,\;1.9$ and 3.9 m/s, respectively. The roughness Reynolds number ($u^{*}z_{0}/\nu$) ranged from 2.91 to 12.23, where $z_{0}$ is the roughness length and ν is the kinematic viscosity. In addition, $u^{*}$ values of 0.04,0.06,0.07,0.15 m/s, and a roughness Reynolds number between 1.53 and 4.40 were evaluated for the roughness configuration. The magnitudes of these roughness Reynolds numbers are consistent with the physical considerations presented and discussed by Robins et al. [31,32].

Previous tests were carried out to evaluate the flow temperature stability during the experiments. Temperature stationarity could be verified due to the cooling system external to the wind tunnel that allows the flow temperature to be kept constant outside the boundary layer. This can be seen in the results of the temperature profiles where there are no temperature inversions in the upper part of the profile.

### 2.2. The Hilbert–Huang Spectral Analysis

The turbulence energy spectrum represents the amount of turbulent kinetic energy (TKE) associated with eddies of different scales. In this study, the spectral analysis was performed using the Hilbert–Huang transform method, proposed by Huang and coworkers [33]. The Hilbert–Huang transform method is an adaptive spectral analysis based on the local characteristics of the signal, and is therefore sensitive to transients and non-linearities.

This spectral analysis is based on a preprocessing step that separates a signal into so-called intrinsic mode functions (IMFs), such as singular value decomposition. Upon applying the Hilbert transform to each mode, the instantaneous frequency spectrum of each of the components is obtained. Determining IMFs is challenging in practical applications; thus, a mathematical method to determine the IMFs is replaced by an empirical decomposition method, giving birth to the Hilbert–Huang transform.

The Hilbert–Huang transform is composed of two parts: (a) the empirical mode decomposition (EMD), which splits signals into a small number of IMFs which represent the natural modes of oscillations of the signal; (b) the Hilbert spectral analysis, which provides the energy distribution in the time–frequency domain.

In general, intrinsic mode functions are defined by functions with the following properties that provide a well-behaved Hilbert transform:1.The number of local extrema and zero-crossings may differ by a maximum of one;2.The mean value of the envelope defined by the local maxima and the envelope defined by the local minima is zero.

As previously mentioned, the Hilbert spectral analysis examines the instantaneous frequency of each intrinsic mode function and its time evolution. Therefore, the result is a frequency-time distribution of a signal amplitude that may reveal localized features manifested in variable amplitudes and frequencies with time. The intrinsic mode functions are determined through a process called sifting, with the following steps:1.Identify all the local extrema in the data sample u(t);2.Parameterize the local maxima/minima by a cubic spline line as the upper/lower envelope;3.Evaluate the mean $m_{1}$ between the upper and lower envelopes;4.Determine the first protomode
(1)$$h_{1}=u\left(t\right)-m_{1}.$$

After the first sifting process, new extrema are generated and reveal the modes lost in the initial examination. A subsequent sifting process is now applied on $h_{1}$ to refine the first protomode
(2)$$h_{1,1}=h_{1}-m_{1,1}.$$

The subsequent sifting steps are generically given for the *k*-th protomode by
(3)$$h_{1,k}=h_{1,(k-1)}-m_{1,k}$$
which is signed as an intrinsic function mode $C_{1}=h_{1}$ if a certain stoppage criteria is achieved.

The stoppage criteria used is the convergence test [33,34]
(4)$$\frac{\sum_{t=0}^{T}{\left|h_{1,(k-1)}\left(t\right)-h_{1,k}\left(t\right)\right|}^{2}}{\sum_{t=0}^{T}{\left[h_{1,(k-1)}\left(t\right)\right]}^{2}}<\epsilon \;,$$
where ϵ is a pre-defined value.

By design, $C_{1}$ should contain the smallest scale of the signal. The subsequent intrinsic mode functions are determined upon subtracting the first IMF $C_{1}$ from the sample $u\left(t\right)-C_{1}=r_{1}$. The first residual $r_{1}$, may still contain important information (i.e., longer scales), and it is treated as the new data input and subjected to the same sifting process as for $C_{1}$ and generically $r_{n-1}-C_{n}=r_{n}$. The decomposition ends when no more intrinsic mode functions can be extracted from $r_{n}$. The original signal can be represented as
(5)$$u\left(t\right)=\sum_{j=1}^{n}C_{j}+r_{n}.$$

From the intrinsic mode function components, the instantaneous frequency can be determined by applying the Hilbert transform of each intrinsic mode function
(6)$$\hat{C}\left(t\right)=\frac{1}{\pi }P\int_{-\infty }^{\infty }\frac{C(\tau )}{t-\tau }d\tau ,$$
where *P* is the Cauchy principal value. With the pair of Hilbert transform C(t) and $\hat{C}\left(t\right)$, the analytical function z(t) is obtained by
(7)$$z\left(t\right)=C\left(t\right)+i\hat{C}\left(t\right)=a\left(t\right)e^{i\theta (t)},$$
where $i=\sqrt{-1}$, $a\left(t\right)=\sqrt{C^{2}+{\hat{C}}^{2}}$ and $\theta \left(t\right)=\tan^{-1}\left(C/\hat{C}\right)$ are, respectively, the instantaneous amplitude and phase of *C*. In this context, the instantaneous frequency can be defined as
(8)$$f=\frac{1}{2\pi }\frac{d\theta }{dt}.$$

The original input data can be expressed by the following expression:(9)$$u\left(t\right)=\sum_{j=1}^{n}a_{j}\left(t\right)e^{i2\pi \int f_{j}\left(t^{\prime }\right)\;dt^{\prime }}.$$

The distribution of the energy of u(t) (square of the amplitude *a*) in the time and frequency spaces is called the Hilbert–Huang spectra (H(f,t)). The Hilbert–Huang marginal spectra h(f) is defined by
(10)$$h\left(f\right)=\int_{0}^{T}H\left(f,t\right)dt,$$
where *T* is the sampled period.

As can be seen in Equation (Equation 10), the Hilbert–Huang marginal spectrum provides the same information than the classical Fourier energy spectrum. Nonetheless, in the Hilbert–Huang transform, the frequencies can be represented in the limit of the continuum, unlike the Fourier transform in which the discrete frequencies are integer multiples of $T^{-1}$. In this analysis, the frequencies on the Hilbert–Huang marginal spectra are divided on 256 logarithmically spaced points.

Figure 2 compares the power spectrum densities obtained using both Fourier and Hilbert–Huang transforms. The classical Fourier spectrum presents a large variability due to spurious harmonics. To overcome this limitation a block-average method is used in which an averaged spectra is obtained from the spectra evaluated for different non overlapping segments of the original signal. This procedure reduces the range of frequencies represented by the spectrum. In the example presented (Figure 2), the anemometric signal is divided into 44 blocks of 8192 values. As consequence, all the energy associated with the frequencies lesser than 0.3 Hz was lost. On the other hand, the Hilbert–Huang marginal spectrum is very smooth in the higher-frequency region and provides a better representation of the energy associated to the lower frequencies than the Fourier spectrum. For easy identification of the energy peak on the low-frequency regions, a moving average process is applied to the Hilbert–Huang marginal spectrum.

## 3. Experimental Results

The height of the boundary layer was approximately 150 mm with a smooth wind tunnel floor. When roughness elements were inserted onto the test chamber floor upstream of the M-II turntable, the height of the boundary layer increased beyond 300 mm. Each roughness element is a prism with a square section (20 mm × 20 mm) and 30 mm in height. The roughness distribution on the floor surface is defined by a 160 mm × 160 mm rhombus where an element is located at each vertex. Roughness is distributed over the entire width of the test section covering 7 m in length upwind of the test table (Figure 3).

### Temperature and Velocity Profiles

The stability regimes are related to the vertical temperature profile shown in Figure 4 for $U_{\infty }=0.9,\;1.3,\;1.9$ and 3.9 m s${}^{-1}$. In unstable cases with and without surface roughness, a pronounced negative temperature gradient was observed due to heating the turntable M–II surface.

For turbulence parameterizations, as proposed by Degrazia et al. [35] one of the most important variables is the vertical profile of the longitudinal velocity *U*, which was measured for $U_{\infty }=0.9,1.3,1.9$ and 3.9 m s${}^{-1}$ for neutral and convective stability regimes with and without surface roughness. The profiles for these four velocities are shown in Figure 5. The surface roughness led the respective curves to have lower mean velocities for specific heights than the smooth wind tunnel floor values. The wind speed profile in the neutral boundary layer varies logarithmically. Thus, the wind speed profiles should be linear in respect to the logarithm of the height. This is verified for the neutral profiles in the boundary layer region (Figure 5). In such cases, the correlation coefficients between the linear fits (not shown) and the evaluated wind speed profiles ranges from 0.99 to 1.00. In the unstable conditions, the wind speed vertical profiles are deflected from the linear curve due the influence of the vertical heat flux. As the $U_{\infty }$ increases, the influence of the convective forcings becomes less effective and the wind speed profiles resemble the neutral shape (Figure 5d).

In summary, analyzing the behavior of the longitudinal velocity and temperature profiles developed with the surface roughness of the wind tunnel showed that the turbulence developed by this constraint in the superficial velocity is responsible for the more efficient vertical transport of heat and momentum. Therefore, a superficial roughness prevents, at low heights, the homogeneous formation of vertical profiles.

Turbulent velocity fluctuations are connected to the turbulence energy spectrum [29]. Turbulence intensities were evaluated from the data for $U_{\infty }=0.9,1.3,1.9$ and 3.9 m s${}^{-1}$ for neutral and convective stability with and without surface roughness (Figure 6).

Turbulent velocity fluctuations and turbulent velocity standard deviations are connected to the turbulence energy spectrum [36]. These fluctuations represent quantities that allow turbulence intensity estimation and were evaluated from the data for $U_{\infty }=0.9,1.3,1.9$ and 3.9 m s${}^{-1}$ for neutral and convective stability with and without surface roughness (Figure 6). For low heights (up to 50 mm), the fluctuations are clearly influenced by the heat flux. These results show that the convective thermal forcing amplifies the turbulent velocity fluctuations for the different magnitudes of the free flow velocity in the vicinity of the surface. The depth or vertical extent of the boundary layer is defined as the thickness of the turbulent region next to the surface [3]. This is also called the depth of the mixed layer since scalar and vector quantities, due to turbulent mixing, are well mixed within it. Therefore with the growth of the free flow velocity in the wind tunnel with a rough surface (neutral or unstable), the gradient of the turbulent intensity decreases. The mixing caused by the vertically extended turbulence is responsible for this decrease [37]. However, without surface roughness and the increased free flow velocity, the gradient of the turbulent intensity to the neutral and unstable stability shows similar magnitudes. Without the rough surface turbulent forcing, the vertical extent of turbulence decreases causing a more pronounced turbulent intensity gradient.

The gradient Richardson number $Ri_{g}$ is used to compute the ratio between convective effects against inertia [7]. This non-dimensional number is commonly used to analyze thermally stratified flows in wind tunnel and it is defined by
(11)$$Ri_{g}=\frac{g\Delta T\Delta Z}{T\Delta U^{2}}$$
where *g* is the gravitational acceleration, *T* is a reference temperature, ΔT is the temperature difference and ΔU is the the mean wind speed difference between two vertical levels separated by ΔZ. Table 1 shows the values of the $Ri_{g}$ number obtained with the different test mean velocities for both cases, smooth surface and rough surface. Convective effects decrease versus inertial effects when mean velocity is increased. This behavior is verified by the decrease in Ri in both cases, but is more pronounced in the case of smooth surface.

#### Spectral Analysis Results

The Hilbert–Huang spectra for the heights *z* = 5, 25, 70 and 150 mm, where the top line is for $U_{\infty }=0.9$ m s${}^{-1}$, the middle line is for 1.9 m s${}^{-1}$ and the bottom line is for 3.9 m s${}^{-1}$ are shown in Figure 7. For lower heights (z≤50 mm), the shapes of the spectra are comparable, whereas the spectra are dominated by the presence or absence of the surface roughness at higher heights (z≥70 mm). The smooth surface spectra contain significantly less energy than the ones with surface roughness, which is observable for z=150 mm. One of the reasons is that without surface roughness, the boundary layer height is approximately 100 mm, while in the case with surface roughness, it is beyond 300 mm. In regions very close to the surface, regardless of velocity and roughness, convective spectra exibit a higher energy content when compared to the neutral spectra. However, in this vertical region, comparing smooth and rough surfaces no spectral changes are observed. These experimental observations agree with the results of the numerical simulations developed by Lee et al. [38].

The normalized spectra are presented for a convective regime with a smooth wind tunnel floor (Figure 8). The normalized spectra for velocity $U_{\infty }=1.3$ m s${}^{-1}$ are compared for the four heights z=5,15,25,100 mm (Figure 8a). There are two spectral maxima for all heights, with the dominant maximum located at higher frequencies, which is due to eddies of mechanical origin, and a secondary maximum located at lower frequencies due to eddies originating from convection. These properties of the two maxima are also identified in observational data of the convective planetary boundary layer [39,40].

In the case of the unstable atmospheric boundary layer, the large eddies have their length scale described in terms of the height of the first upper inversion which occurs in higher altitudes. Differently, in the wind tunnel simulations, the effect of the spectral maxima for a convective stability regime are smaller compared to those associated to turbulence from mechanical origin and with higher frequencies. Moreover, its intensities decrease with increasing height. Additionally, Figure 8 indicates with dashed line the region where Kolmogorov’s law applies, i.e., in the inertial sub-interval.

The normalized spectra obtained when the height was chosen as z=10 mm and the free flow velocity influence is shown in Figure 8b for $U_{\infty }=0.9,1.3,1.9$ and 3.9 m s${}^{-1}$. By increasing the velocity, the intensity of the spectral maxima associated with low frequencies and convective turbulence decreases. Nonetheless, with increasing free flow velocity, the spectral maxima associated with mechanical turbulence in the range of higher frequencies shifted towards higher values. Figure 9 shows the corresponding plots for Figure 8 but with surface roughness, where similar analyses can be made.

## 4. Conclusions

This study showed wind tunnel experiments designed to mimic typical properties in the planetary boundary layer. Thermal effects caused by sunlight were simulated using Peltier elements and were responsible for heating the floor of the wind tunnel. This heating of the surface of the wind tunnel generated typical vertical temperature profiles of unstable boundary layers. Analysis of these temperature profiles revealed that as the free flow velocity increased, the vertical temperature gradient weakened. This decrease in the temperature gradient is more evident in the absence of surface roughness. The results show that the presence of surface roughness generates a deeper turbulent boundary layer. The presence of a turbulent environment filling higher vertical regions means a deepening of the boundary layer. In this case, there is an exchange process of momentum involving greater heights in the boundary layer. However, when there is no roughness on the surface of the wind tunnel, the vertical temperature and longitudinal velocity profiles become approximately constant at lower heights than those with a rough surface. Therefore, the mechanical turbulence activated by the roughness promotes the exchange of momentum and heat between farther vertical regions of the wind tunnel boundary layer.

The results of this study show that in the vicinity of the surface, for the different magnitudes of the free flow velocity, fluctuations in turbulent velocity are significantly influenced by the convective forcing. In the presence of a heat turbulent flux in regions neighboring the surface, thermal effects enhance the intensity of turbulence. With the growth of the free flow velocity in the wind tunnel with a rough surface (neutral or unstable case), the gradient of the turbulent velocity standard deviation decreases. The turbulent mixing caused by the vertically extended turbulence is the mechanism responsible for this decrease. However, without the presence of surface roughness and with the increase of the free flow velocity, the gradient of the turbulent velocity standard deviation to the neutral and unstable stability exhibits coincident magnitudes. Without the surface roughness, limited vertically turbulence (reduced turbulent boundary layer depth) establishes and consolidates a stronger gradient for the turbulent velocity standard deviation.

Regarding the Hilbert–Huang spectra, the results indicate that the convective forcing generating the turbulence energy dominates the mechanical forcing in regions near the surface, regardless of the free flow velocity. This experimental fact corroborates the observed behavior of the turbulent velocity standard deviation. For greater magnitudes of free flow velocity and higher heights, the turbulence energy in the spectra is controlled by the surface roughness. This amplified mechanical turbulence is responsible for the transport and consequent homogenization of the turbulent statistics. The presence of thermal effects in the generation of turbulent convective energy is revealed by the manifestation of small spectral peaks located in low-frequency windows in the turbulence energy spectrum. These spectral peaks present low intensities and originated from the buoyancy forcing mechanism decrease with increasing height and free flow velocity. However, the turbulence spectrum manifests dominant spectral peaks associated with the energy generated by mechanical effects at higher frequencies. With higher free flow velocity, these spectral peaks originated by the wind shear forcing move to higher frequencies. This pattern for the turbulence energy spectra, with dominant and secondary spectral peaks, is also observed in the horizontal velocity components in an unstable surface planetary boundary layer.

## Figures and Tables

**Figure 1 ijerph-19-05134-f001:**
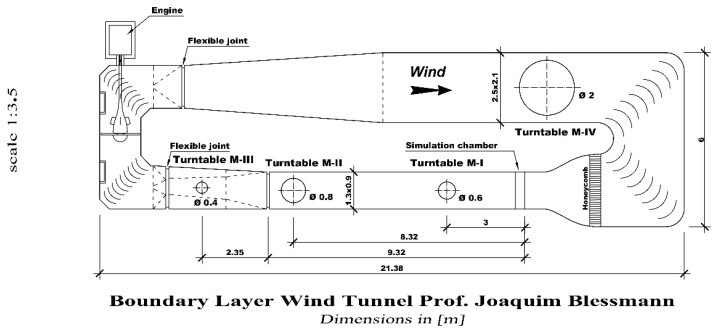
Wind tunnel “Prof. Joaquim Blessmann”, Laboratório de Aerodinâmica das Construções–Universidade Federal do Rio Grande do Sul (LAC–UFRGS).

**Figure 2 ijerph-19-05134-f002:**
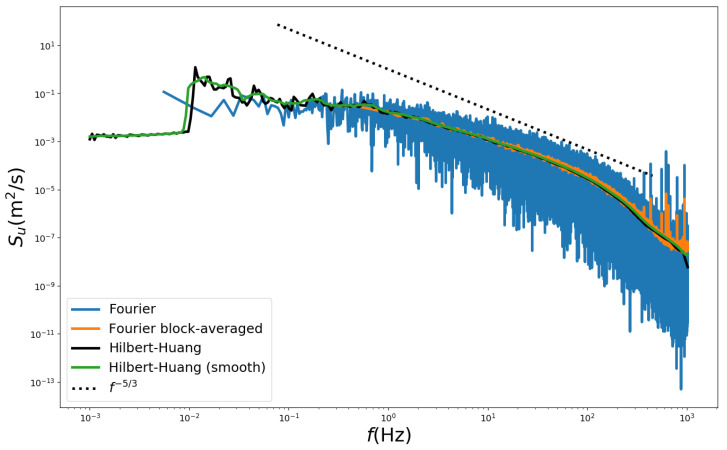
Power spectra density of velocity fluctuations for neutral flow at *z* = 30 mm ($U_{\infty }$ = 3.9 m/s) evaluated using Fourier and Hilbert–Huang transform. Doted line represents the Kolmogorov −5/3 law.

**Figure 3 ijerph-19-05134-f003:**
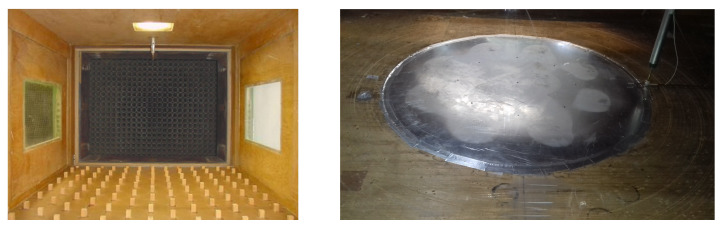
Roughness characteristics in the wind tunnel test section (**left panel**) and experimental section with Peltier elements (**right panel**).

**Figure 4 ijerph-19-05134-f004:**
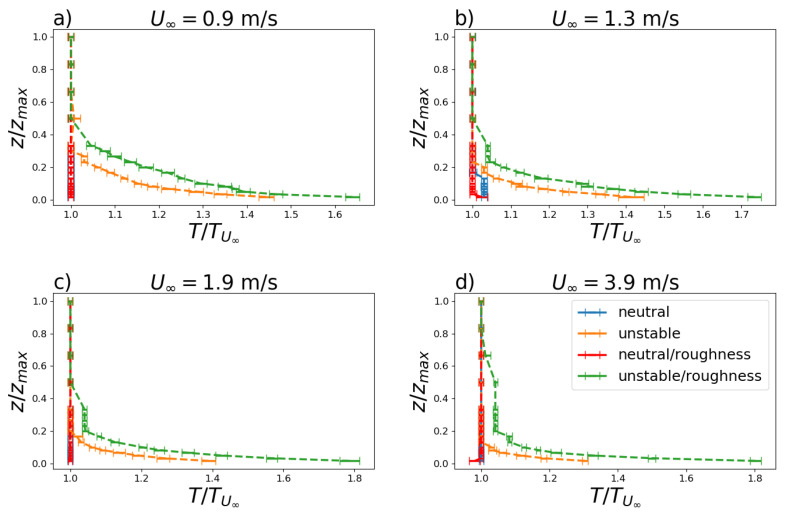
Vertical temperature profile for $U_{\infty }=0.9,\;1.3,\;1.9$ and 3.9 m s${}^{-1}$ (**a**–**d**) for neutral and convective stability (unstable) with (roughness) and without surface roughness. Horizontal lines represent the standard deviations.

**Figure 5 ijerph-19-05134-f005:**
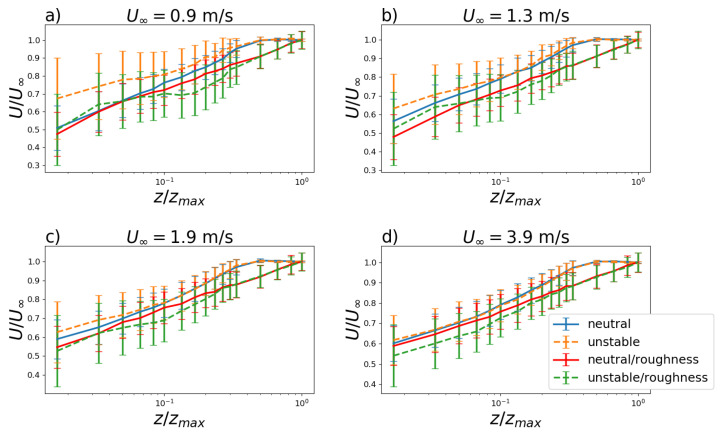
Vertical profile of the longitudinal mean velocity *U* for $U_{\infty }=0.9,\;1.3,\;1.9$ and 3.9 m s${}^{-1}$ (**a**–**d**) for neutral and convective stability with and without surface roughness. Vertical lines represent the turbulent velocity standard deviations.

**Figure 6 ijerph-19-05134-f006:**
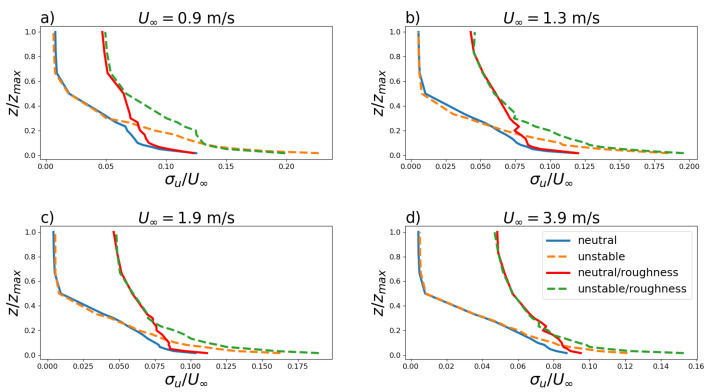
Vertical profile of turbulence intensity $\left(\sigma_{u}/U_{\infty }\right)$ for $U_{\infty }=0.9,\;1.3,\;1.9,\;3.9$ m s${}^{-1}$ (**a**–**d**) for neutral and convective stability with and without surface roughness.

**Figure 7 ijerph-19-05134-f007:**
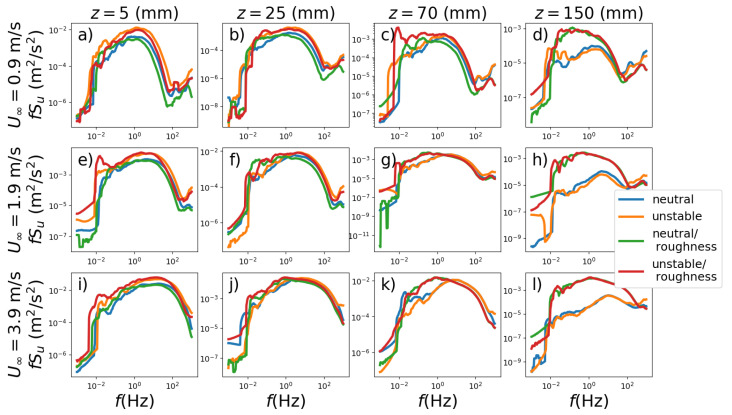
The Hilbert spectra (z=5,25,70 and 150 mm, respectively) are shown in the vertical panels. The $U_{\infty }=0.9,\;1.9$ and 3.9 m s${}^{-1}$ are in the horizontal panels, respectively. The (**a**–**l**) exhibit different heights and speeds.

**Figure 8 ijerph-19-05134-f008:**
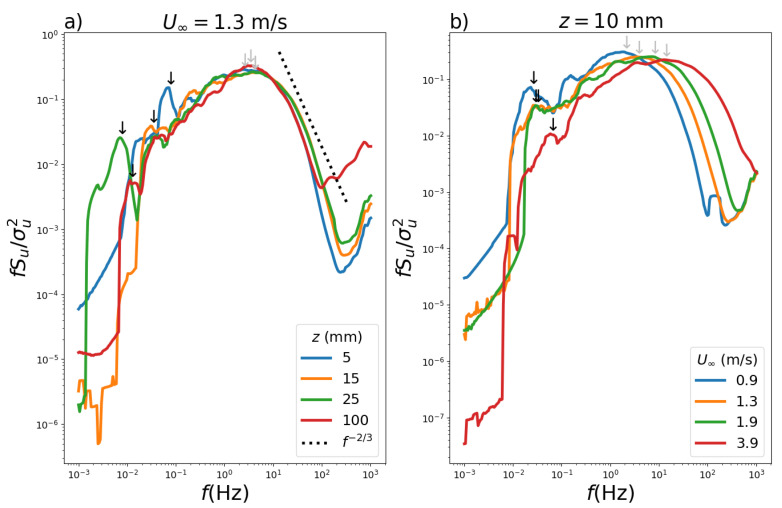
Normalized Hilbert–Huang spectrum (**a**) $U_{\infty }=1.3$ m s${}^{-1}$ (without roughness) for heights z=5,15,25,100 mm and (**b**) with z=10 mm for $U_{\infty }=0.9,\;1.3,\;1.9$ and 3.9 m s${}^{-1}$. The arrows point the dominant (gray) and secondary (black) peaks.

**Figure 9 ijerph-19-05134-f009:**
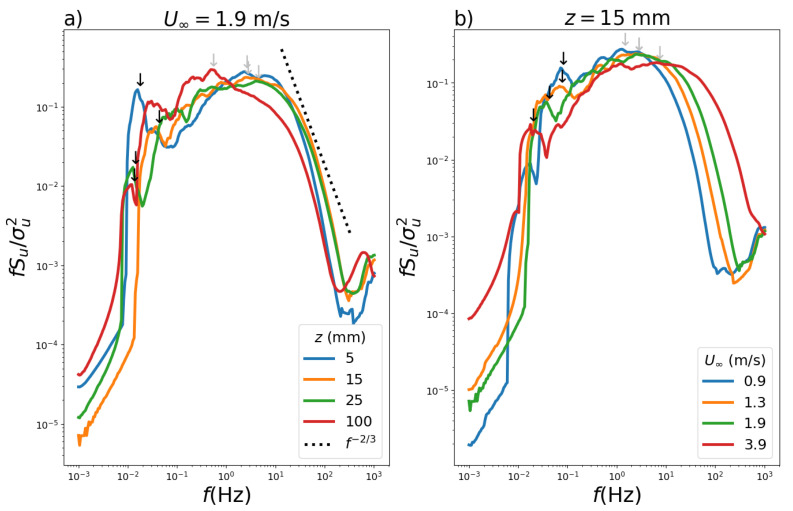
Normalized Hilbert-Huang spectrum (**a**) with $U_{\infty }=1.9$ m s${}^{-1}$ (with roughness) for heights z=5,15,25 and 100 mm and (**b**) with z=15 mm for $U_{\infty }=0.9,\;1.3,\;1.9$ and 3.9 m s${}^{-1}$. The arrows point the dominant (gray) and secondary (black) peaks.

**Table 1 ijerph-19-05134-t001:** Gradient Richardson number for smooth and rough surface.

	Smooth Surface				Roughness Surface			
Mean velocity [m s${}^{-1}$]	0.9	1.3	1.9	3.9	0.9	1.3	1.9	3.9
ΔT [K]	−8.5	−8	−7	−6	−9	−11	−11.5	−11.5
ΔU [m s${}^{-1}$]	0.3	0.45	0.75	1.53	0.41	0.59	0.78	1.6
$Ri_{g}$	−0.459	−0.192	−0.06	−0.012	−0.539	−0.318	−0.19	−0.045

## Data Availability

The data used in this study are available by contacting the corresponding author.

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
