# Peer review of "Analysis of Thermal and Roughness Effects on the Turbulent Characteristics of Experimentally Simulated Boundary Layers in a Wind Tunnel"

_ijerph, 2022, doi:10.3390/ijerph19095134_

Round 1

Reviewer 1 Report

This paper develops on the wide area of research in atmospheric boundary layer modelling by including thermal effects. Solar radiation fluxes were modelled using Peltier elements on the wind tunnel floor. Comparisons were also made between a smooth wall and rough wall ahead of the test section. It was found that the vertical temperature gradient weakened as velocity increased, the roughness elements generate a deeper turbulent boundary layer and the roughness promotes a larger vertical exchange of heat and momentum. Overall, I was impressed with the work and think it is a valuable contribution to the research community working on modelling ABL flows in wind tunnels. However, I have some comments that I think should be addressed:

1) Firstly, one of the most important parameters for characterizing a boundary layer flow is the friction velocity, sometimes called u_tau or u*. It is important that the authors estimate this value to fully characterise the flow. 

2) The errors bars on the velocity measurements in Figure 5 are particularly large. Because of this, the results are not particularly meaningful and do not necessarily prove that the mean velocity follows a logarithmic profile. This should be addressed.

3) It would be useful to include a more detailed dimensioned schematic of the wind tunnel, the roughness, the Peltier setup, and the measurement locations.

4) How was the roughness geometry chosen and how does that choice impact the boundary layer? How did you measure the increased boundary layer thickness? Are there other specific impacts as well?

5) Finally there are several minor grammatical mistakes that should be corrected. Here are some that I noted:

  • L1: is to analyse 
  • L28: . The larger the eddies, the more horizontal
  • L36: considering these arguments
  • L38: may not be able to include effects 
  • L87: In this study both neutral and convective wind tunnel boundary layers are considered.
  • L177: the stoppage criteria used is the convergence test
  • Table 1: "Smooth surface"

Reviewer 2 Report

This article presents a series of hot wire measurements of a turbulent boundary layer, with or without thermal effects and surface roughness. The end goal is the simulation of the flow-physics of the planetary boundary layer in a wind tunnel. While the measurements and experimental apparatus seem to be reliable, from a fluid-mechanics perspective the article is incomplete.

I list some key reasons below

The flow properties are not non-dimensionalized. For instance, I could not find the Reynolds related to the tau or momentum thickness.  What is the relative size of the roughness element relative to the viscous sublayer?

How did you measure the temperature?

The authors barely mention any previous work on thermal boundary layers. They just state that “generally” this is not studied (line 47). This is not true, many previous works exist in the literature, for instance Doosttalab et al. JFM (2016) and references therein. A literature review is actually necessary. How can one hope to assess if your findings have not been reported in previous works?

Several of the statements in the article are trivial, for instance line 216, the log-region in a boundary layer is known for almost 100 years. That turbulence fluctuations directly allow the calculation of the turbulence intensity via the latter’s definition (line 231) is also trivial.

Other comments.

Line 27. The planetary boundary layer has a 2D structure etc. Please cite.

Line 31. Stationarity of the Navier-Stokes. You mean Kolmogorov’s equilibrium assumption? I do not understand your sentence.

Line 268. I do not understand what you mean.

Round 2

Reviewer 1 Report

I am happy with the revisions.

Reviewer 2 Report

The manuscript has been significantly improved. However, there are still two points, one major and one minor, which require revision.

1) It is not enough to cite two older works on thermal and rough boundary layers. It is necessary to mention how your results add to the existing literature, especially given that your conclusions are, to a certain extent, qualitative. One gets the impression that what you present may have already been found, presented and discussed.

2) The following expression seems inaccurate to me "For this type of turbulence, for large times, the solution of the Navier-Stokes equation tends to a stationary state with a finite mean energy per unit mass.

Navier Stokes does not tend to a stationary solution (there are continuous fluctuations in turbulence). If the authors want to include this information I would rephrase it, e.g. Small scale turbulence, far from initial conditions and at high Reynolds numbers, tends to reach a statistically stationary state. An article that discusses Kolmogorov's assumption is: K. Steiros (2022) Balanced nonstationary turbulence. Physical Review E.
